# Pan Trapping and Malaise Trapping: A Comparison of Bee Collection Techniques in Subalpine Meadows

Nicholas Anderson [1], Steven Petersen [1], Robert Johnson [2], Tyson Terry [3], Jacqueline Kunzelman [1], David Lariviere [1] and Val Anderson [1,*]

1 Department of Plant and Wildlife Sciences, Brigham Young University, Provo, UT 84602, USA; nva22@byu.edu (N.A.); steven_petersen@byu.edu (S.P.)
2 Department of Biology, Brigham Young University, Provo, UT 84602, USA; robert_johnson@byu.edu
3 Disturbance Ecology Department, University of Bayreuth, 95444 Bayreuth, Germany
* Correspondence: val_anderson@byu.edu

**Abstract:** Public lands, managed for multiple uses such as logging, mining, grazing, and recreation, also support vital environmental services like wild bee pollination. A trending decline in wild bees has heightened interest in documenting these key pollinators in their native habitats. Accurate assessment of pollinator community diversity is crucial for population monitoring and informing land management practices. In this study, we evaluate the efficiency of Malaise traps and pan traps in sampling wild bees over three growing seasons in subalpine meadow communities in central Utah. Sixteen trapping sites were established, each with a Malaise trap and an array of blue, white, and yellow pan traps, nine at each site. Weekly collections were made through summer months and a comparison of their effectiveness in capturing bee abundance and species richness was made. Malaise traps captured significantly greater abundance of bees on average, though this was species-dependent. Malaise traps were especially effective at capturing *Bombus* spp. and larger species. Pan traps were generally more effective with smaller species such as *Hylaeus* spp. White pan traps outperformed yellow and blue pan traps in terms of abundance and only yellow pan traps in terms of richness. Both methods contributed unique species to the overall collection effort, suggesting that a combination of trapping methods provides a more comprehensive understanding of bee communities. Species accumulation curves indicate that species existing within the community went unencountered in our samples and that more time or perhaps additional methods could aid in best describing the entire community.

**Keywords:** pollinators; insects; monitoring; methods; population; diversity; abundance; entomology

## 1. Introduction

Much of our public lands are managed for multiple uses including logging, mining, grazing, recreation, wildlife habitat, and others. Some uses are less evident and are related to environmental services provided by healthy ecosystems. The pollination of plants by wild bees is one such service that has far reaching implications. Evidence of wild bee decline has fueled interest over the past few decades in understanding and documenting these key pollinators in their native habitats [1–11]. Pollinator health, specifically that of bees, is a key concern for state and federal agencies on public lands [12]. The ability to accurately assess trends in pollinator community diversity can help inform land managers regarding management practices that will sustain healthy and stable pollinator communities. Different capture techniques have been documented for inventorying wild bees including aerial and sweep netting [13,14], vane traps [15], sticky traps [16], pan traps [17,18], Malaise traps [19], and transect observation [20].

Many methods for biological inventories rely heavily on in-person sampling. Carril et al. [21] conducted one of the largest-scale bee inventories in North America throughout

the Grand Staircase-Escalante National Monument in southern Utah. Over the course of 1632 collector days, this study resulted in the documentation of 660 species, several new state species records, and the discovery of 49 species previously unknown to science. This effort was conducted primarily with the use of aerial netting, supplemented by pan traps. Large-scale active sampling (i.e., hand netting) is time- and labor-intensive. It is also subject to sample bias by favoring the capture of larger, slower, and less discrete bee species that are active during collection hours as well as variability in observer experience [22]. Some such biases were illustrated by Lindström et al., [23] who reported that 94% of observed individuals across sample transects were "impossible" to catch by hand netting and were not identifiable beyond the family level in situ. Active, in-person sampling does provide contextual and spatial benefits such as the ability to document associated forage plant species and the ability to distribute sample effort across the assemblage of flowering plant species.

An alternative to hand netting is to passively sample using traps. Pan and Malaise traps are commonly used to sample wild bees and have the advantage of collecting continually and consistently over an extended period of time with minimal field effort. These sampling methods, however, can be disturbed or disabled by inclement weather, livestock, or other wildlife and they often collect a large amount of non-target insects, which increases sorting time to isolate species of interest. Every insect sampling method has advantages, disadvantages, and capture biases that can erroneously influence assumptions. Some have suggested that a combination of collection methods may be needed to overcome these biases and accurately assess native bee diversity [14]. Studies exploring how different sampling techniques perform, particularly in different environments, provide both a deeper understanding of local insect communities and insight that can help researchers apply techniques appropriately to collect the data they require.

The purpose of this study was to assess the strengths and limitations of different sampling techniques for collecting bee specimens and quantifying pollinator community structure. To accomplish this, we collected, identified, and analyzed insect samples across three consecutive years during the growing season using two different passive sampling techniques: Malaise traps and pan traps. We observe and discuss the advantages, disadvantages, and efficiency associated with these passive trapping methods as well as the differences in capture rates and resulting bee richness. These differences are observed on a community scale and assessed at the level of subfamily, genus, and species where appropriate.

## 2. Materials and Methods

### 2.1. Study Design

Sixteen sites representing the subalpine meadow community across the central Wasatch Plateau, Sanpete County, Utah, in the Manti-La Sal National Forest were selected for this study (Figure 1). An array of nine pan traps and one Malaise trap were deployed at each site over the course of three growing seasons, 2017, 2018, and 2019, in open subalpine meadows. Each site was distanced greater than 0.80 km from any other site to establish independence. Due to seasonal accessibility, heavy grazing presence, wildlife disturbance, or human interruption, not all sites were replicated each year of this study. Daytime temperatures ranged between 5 and 25 degrees Celsius during the collection window. Sites were located between elevations of 2900 and 3200 m above mean sea level. Precipitation is delivered in the form of heavy snowpack in the winter and sporadic monsoonal rain events in summer months. A baseline vegetation inventory was made at each location and a table of common flowering species at trapping sites is included (Table 1).

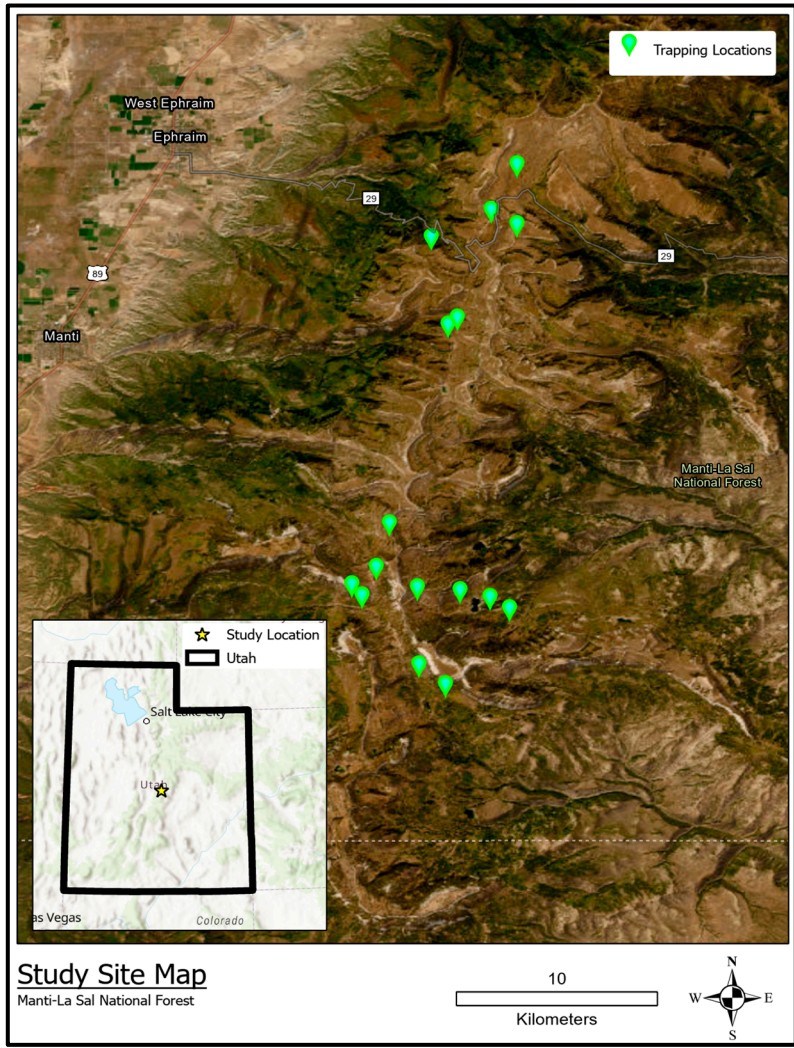

**Figure 1.** Map featuring locations of each trapping site.

Trapping season on these high meadows was limited to between early July when snow melt allowed full road access and early September when the flowering bloom had finished with hard frosts. Townes-style Malaise traps [24] were set up in early July with collection bottles filled with 70% ethyl alcohol. The sample bottles were collected and replaced each week from each Malaise trap until early September. A one-week trapping interval was the maximum time Malaise traps could be reasonably deployed to avoid bottles filling up with insects and/or experiencing excessive evaporative loss to the ethanol. The Townes-style Malaise traps were purchased from Santee Traps. The dimensions of these traps were approximately 2 m long × 1 m wide × 2 m tall at their peak (Figure 2).

Pan traps were deployed over a 24 h period once each week from early July to early September. A 24 h interval was the maximum time pan traps could be reasonably deployed to avoid unexpected weather events or wildlife destroying the traps. Pan traps were 15-ounce plastic bowls with a 17 cm diameter and 7.5 cm depth. They consisted of three colors, blue, white, and yellow, which was consistent with the primary colors of the floral bloom. Pans were elevated up to 45 cm on wooden stakes to reflect surrounding vegetation height and filled with a scentless dish soap water solution (Figure 2). The pan trap array consisted of three transects emanating from a central point. One trap of each color was placed in each transect and spaced every five meters for a total of 9 pans at each site. Colors were placed sequentially so that each color was represented in each position along the array, front, middle, and end.

**Table 1.** Common flowering forb species found in the subalpine meadow communities where Malaise and pan traps were deployed to collect bees in the summer months of 2017, 2018, and 2019.

| Scientific Name | Common Name |
| --- | --- |
| *Achillea millefolium* Linnaeus, 1754 | common yarrow |
| *Artemisia ludoviciana* Nuttall, 1818 | cudweed sagewort |
| *Aster ascendens* Lindley, 1834 | Rocky Mountain aster |
| *Collomia linearis* Nuttall, 1818 | slenderleaf collomia |
| *Delphinium nuttallianum* Pritzel, 1843 | Nutall's larkspur |
| *Erigeron speciosus* (Lindley, 1836) | aspen fleabane |
| *Fragaria virginiana* Duchesne, 1766 | wild strawberry |
| *Geranium viscosissimum* Fisher and Myer 1846 | sticky purple geranium |
| *Helianthella uniflora* Nuttall, 1842 | oneflower helianthella |
| *Lathyrus lanszwertii* Kellogg, 1861 | Nevada sweet pea |
| *Ligusticum porteri* Coulter and Rose, 1888 | Indian parsley |
| *Lupinus argenteus* Pursh, 1814 | silver-stem lupine |
| *Mertensia ciliata* (James, 1838) | mountain bluebells |
| *Orthocarpus tolmiei* Hooker and Amot, 1839 | Tolmie's owl's-clover |
| *Osmorhiza occidentalis* (Nuttall, 1859) | western sweet cicely |
| *Penstemon rydbergii* A. Nelson, 1898 | Rydberg's penstemon |
| *Polemonium foliosissimum* A. Gray, 1878 | leafy Jacob's ladder |
| *Potentilla gracilis* Douglas, 1830 | slender cinquefoil |
| *Rudbeckia occidentalis* Nuttall, 1841 | western coneflower |
| *Smilacina stellata* Linnaeus, 1753 | starry false Solomon's seal |
| *Solidago multiradiata* Aiton, 1789 | Rocky Mountain goldenrod |
| *Stellaria jamesiana* Torrey, 1828 | James' starwort |
| *Taraxacum officinale* Weber, 1780 | common dandelion |
| *Thalictrum fendleri* Engelmann, 1849 | Fendler's meadow-rue |
| *Vicia americana* Muhlenberg, 1801 | American vetch |
| *Viguiera multiflora* (Nuttall, 1848) | showy goldeneye |
| *Viola purpurea* Kellogg, 1855 | goosefoot violet |

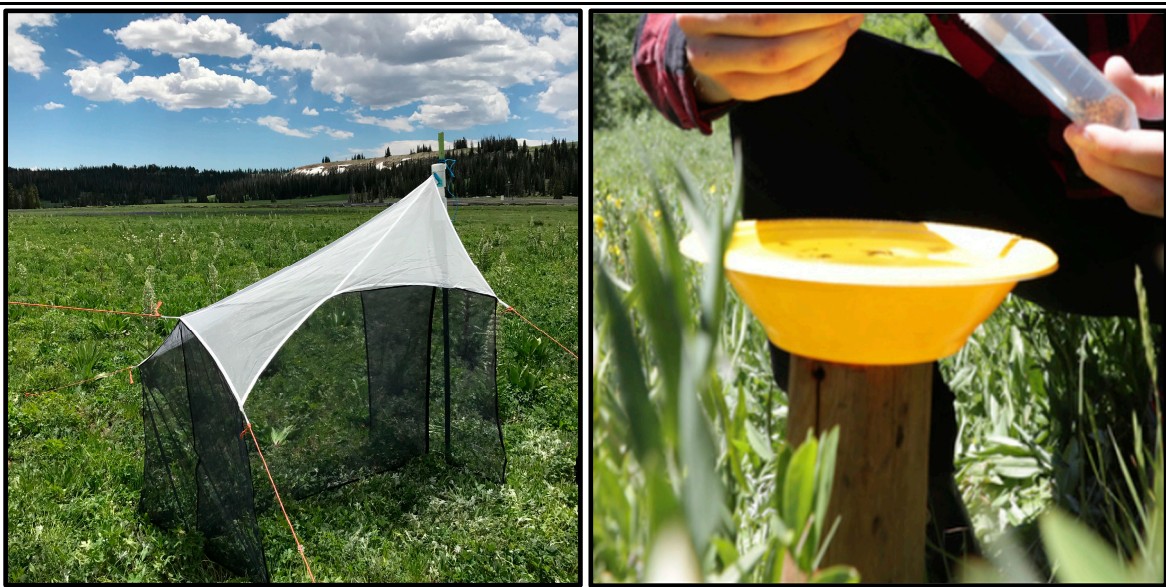

**Figure 2.** A Malaise trap (**left**) and elevated yellow pan trap (**right**) deployed at a study site in the Manti-La Sal National Forest in the early summer of 2018.

The amount of in-field time required to set up and subsequently collect samples from the traps was roughly equivalent for both trapping methods. However, the lab work associated with sorting, processing, and identifying specimens was dependent upon the number of target and non-target insects captured in each individual sample.

## 2.2. Identification

After samples were collected from the field, they were brought to the lab for sorting and the analysis. Bumble bees were identified using the key provided by "Bumble Bees of the Western United States" [25] while all other bees were mounted, labeled, and then delivered to the USDA Pollinating Insect-Biology, Management, Systematics Research unit in Logan, Utah, for identification by bee taxonomic specialists. Bee specimens were identified to the species level when possible. Some taxa were only identified to the genus.

## 2.3. Data Analysis

Each week-long Malaise trap sample was compared to a 24 h pan trap sample deployed in that same week from the same site. Using a mixed model analysis in JMP® software version 17.2.0 [26], measures of annual abundance and community richness were compared between pan and Malaise traps along with an assessment of each color of pan trap. Individual trapping sites were assigned as a random variable and when necessary, $\log (x + 1)$ transformations were utilized as necessary to satisfy the assumptions of the various analyses that were conducted. Metrics of similarity were measured using a PCA analysis in software R 4.3.1 [27], and PrimerE v7 software [28] was utilized to create a species accumulation curve using a variety of different models.

## 3. Results

Over the course of this study, 12,480 wild bees were captured, processed, and identified to the lowest taxonomic level possible. The Malaise traps captured 7912 individual bees and combined pan traps captured 4568 individual bees (Figure 3). These bees represented 119 different species/genera groups from five families. Some species were captured many times and other, rarer, species were only caught once or twice. Both methods caught the same number of these rarer species, Malaise traps captured 19 and pan traps captured 19. On average, a single Malaise trap captured 239 wild bees annually. This was significantly greater than a single array of nine pan traps, which collected an average of 138 bees annually ($p$-value < 0.0001). While overall species richness was similar between the two methods, Malaise traps (101) and pan traps (96), Malaise traps significantly outperformed pan traps in species richness per trap, collecting on average 26 species annually versus 19 ($p$-value < 0.0001). Table 2 provides a complete species list broken down by trap type. Unique species were captured by each method that were not documented in the other. Malaise and combined pan traps had a species richness overlap of 65%. Comparative results varied across the three years, seeing no significant differences in abundance or richness between the two methods in 2017 or 2019, but drastic differences in 2018, when overall capture rates were significantly higher ($p$-value < 0.0001).

**Table 2.** Complete species list with quantities broken down by trap type.

| Bee Species | Total | Malaise Trap | Pan Trap |
|---|---|---|---|
| *Agapostemon femoratus* Crawford, 1901 | 2 | 1 | 1 |
| *Agapostemon texanus* Cresson, 1872 | 1 | 1 | 0 |
| *Andrena apacheorum* Cockerell, 1897 | 23 | 2 | 21 |
| *Andrena birtwelli* Cockerell, 1901 | 13 | 2 | 11 |
| *Andrena cerasifolii* Cockerell, 1896 | 1 | 1 | 0 |
| *Andrena chlorura* Cockerell, 1916 | 42 | 14 | 28 |
| *Andrena colletina* Cockerell, 1906 | 1 | 0 | 1 |
| *Andrena hirticincta* Provancher, 1888 | 26 | 10 | 16 |
| *Andrena laminibucca* Viereck and Cockerell 1914 | 2 | 1 | 1 |
| *Andrena miranda* Smith, 1879 | 4 | 0 | 4 |
| *Andrena surda* Cockerell, 1910 | 1 | 1 | 0 |
| *Andrena thaspii* Graenicher, 1903 | 1 | 1 | 0 |

**Table 2.** *Cont.*

| Bee Species | Total | Malaise Trap | Pan Trap |
|---|---|---|---|
| *Andrena vicinoides* Viereck, 1904 | 18 | 7 | 11 |
| *Andrena* sp. 1 | 3 | 3 | 0 |
| *Andrena* sp. 2 | 1 | 1 | 0 |
| *Andrena* sp. 3 | 124 | 12 | 112 |
| *Andrena* sp. 4 | 1 | 0 | 1 |
| *Andrena* sp. 5 | 1 | 1 | 0 |
| *Andrena* sp. 6 | 2 | 0 | 2 |
| *Andrena* sp. 7 | 1 | 0 | 1 |
| *Andrena* sp. 8 | 1 | 0 | 1 |
| *Anthidium tenuiflorae* Cockerell, 1907 | 3 | 3 | 0 |
| *Anthophora bomboides* Kirby, 1837 | 5 | 5 | 0 |
| *Anthophora terminalis* Cresson, 1869 | 255 | 165 | 90 |
| *Anthophora urbana* Cresson, 1878 | 25 | 24 | 1 |
| *Apis mellifera* Linneaus, 1758 | 36 | 21 | 15 |
| *Ashmeadiella bucconis* Say, 1837 | 1 | 0 | 1 |
| *Ashmeadiella pronitens* (Cockerell, 1906) | 5 | 4 | 1 |
| *Atoposmia* sp. 1 | 1 | 1 | 0 |
| *Bombus appositus* Cresson, 1878 | 745 | 716 | 29 |
| *Bombus bifarius* Cresson 1878 | 1637 | 1582 | 55 |
| *Bombus californicus* Smith 1854 | 69 | 67 | 2 |
| *Bombus fernaladae* Franklin, 1911 | 50 | 50 | 0 |
| *Bombus flavifrons* Greene, 1860 | 1898 | 1839 | 59 |
| *Bombus huntii* Greene, 1860 | 18 | 18 | 0 |
| *Bombus insularis* Nylander, 1848 | 77 | 74 | 3 |
| *Bombus melanopygus* Cresson, 1878 | 1 | 1 | 0 |
| *Bombus mixtus* Cresson, 1878 | 95 | 92 | 3 |
| *Bombus morrisoni* Cresson, 1874 | 1 | 1 | 0 |
| *Bombus nevadensis* Cresson, 1878 | 28 | 27 | 1 |
| *Bombus occidentalis* Greene, 1858 | 40 | 38 | 2 |
| *Bombus rufocinctus* Cresson, 1863 | 198 | 185 | 13 |
| *Bombus sylvicola* Kirby, 1837 | 959 | 935 | 24 |
| *Ceratina* sp. 1 | 1 | 0 | 1 |
| *Coelioxys funeraria* Smith, 1854 | 1 | 1 | 0 |
| *Coelioxys porterae* Cockerell, 1900 | 6 | 3 | 3 |
| *Coelioxys* sp. 1 | 1 | 0 | 1 |
| *Colletes fulgidus* Swenk, 1904 | 23 | 16 | 7 |
| *Colletes hyalinus* Provancher, 1888 | 11 | 10 | 1 |
| *Colletes kincaidii* Cockerell, 1898 | 4 | 4 | 0 |
| *Colletes nigrifron* Titus, 1900 | 4 | 3 | 1 |
| *Colletes paniscus* Viereck, 1903 | 90 | 44 | 46 |
| *Colletes simulans* Cresson, 1868 | 4 | 4 | 0 |
| *Diadasia* sp. 1 | 1 | 0 | 1 |
| *Dufourea harveyi* (Cockerell, 1906) | 31 | 4 | 27 |
| *Epeolus americanus* (Cresson, 1878) | 3 | 2 | 1 |
| *Epeolus minimus* (Robertson, 1902) | 3 | 3 | 0 |
| *Halictus farinosus* Smith, 1853 | 5 | 2 | 3 |
| *Halictus rubicundus* (Christ, 1791) | 23 | 19 | 4 |
| *Halictus tripartitus* Cockerell, 1895 | 1 | 0 | 1 |
| *Halictus virgatellus* Cockerell, 1901 | 45 | 17 | 28 |
| *Heriades cressoni* Michener, 1938 | 13 | 4 | 9 |
| *Hoplitis albifrons* (Kirby, 1837) | 15 | 14 | 1 |
| *Hoplitis clypeata* (Sladen, 1916) | 29 | 10 | 19 |
| *Hoplitis fulgida* (Cresson, 1864) | 53 | 35 | 18 |
| *Hoplitis producta* (Cresson, 1864) | 15 | 4 | 11 |
| *Hylaeus annulatus* (Linneaus, 1758) | 2605 | 172 | 2433 |
| *Hylaeus basalis* (Smith, 1853) | 190 | 42 | 148 |
| *Hylaeus coloradensis* (Cockerell, 1896) | 63 | 6 | 57 |

**Table 2.** *Cont.*

| Bee Species | Total | Malaise Trap | Pan Trap |
|---|---|---|---|
| *Hylaeus* sp. 1 | 40 | 8 | 32 |
| *Hylaeus* sp. 2 | 6 | 0 | 6 |
| *Hylaeus* sp. 3 | 30 | 5 | 25 |
| *Lasioglossum (Dialictus)* spp. | 732 | 554 | 178 |
| *Lasioglossum (Evylaeus)* spp. | 230 | 82 | 148 |
| *Lasioglossum (s. str. sp.)* | 1 | 0 | 1 |
| *Lasioglossum* sp. 1 | 406 | 72 | 334 |
| *Lasioglossum* sp. 2 | 1 | 0 | 1 |
| *Lasioglossum trizonatum* (Cresson, 1874) | 2 | 1 | 1 |
| *Megachile fidelis* Cresson, 1878 | 2 | 1 | 1 |
| *Megachile frigida* Smith, 1853 | 27 | 25 | 2 |
| *Megachile inermis* Provancher, 1888 | 23 | 14 | 9 |
| *Megachile melanophaea* Smith, 1853 | 63 | 59 | 4 |
| *Megachile montivaga* Cresson, 1878 | 5 | 1 | 4 |
| *Megachile nevadensis* Cresson, 1879 | 1 | 1 | 0 |
| *Megachile perihirta* Cockerell, 1898 | 171 | 164 | 7 |
| *Megachile pugnata* Say, 1837 | 36 | 29 | 7 |
| *Megachile relativa* Cresson, 1878 | 43 | 38 | 5 |
| *Melissodes hymenoxidis* Cockerell, 1906 | 9 | 8 | 1 |
| *Melissodes* sp. 1 | 21 | 17 | 4 |
| *Nomada* sp. 1 | 1 | 0 | 1 |
| *Nomada* sp. 2 | 1 | 1 | 0 |
| *Osmia albolateralis* Cockerell, 1906 | 10 | 8 | 2 |
| *Osmia brevis* Cresson, 1864 | 11 | 4 | 7 |
| *Osmia bucephala* Cresson, 1864 | 85 | 55 | 30 |
| *Osmia coloradensis* Cresson, 1878 | 2 | 2 | 0 |
| *Osmia ednae* Cockerell, 1897 | 11 | 7 | 4 |
| *Osmia montana* Cresson, 1864 | 5 | 2 | 3 |
| *Osmia paradisica* Sandhouse, 1924 | 107 | 85 | 22 |
| *Osmia pentstemonis* Cockerell, 1906 | 20 | 17 | 3 |
| *Osmia proxima* Cresson, 1864 | 64 | 53 | 11 |
| *Osmia pusilla* Cresson, 1864 | 29 | 24 | 5 |
| *Osmia raritatis* Michener, 1957 | 1 | 0 | 1 |
| *Osmia sculleni* Sandhouse, 1939 | 112 | 89 | 23 |
| *Osmia simillima* Smith, 1853 | 12 | 5 | 7 |
| *Osmia subaustralis* Cockerell, 1900 | 7 | 6 | 1 |
| *Osmia trevoris* Cockerell, 1897 | 6 | 2 | 4 |
| *Osmia tristella* Cockerell, 1911 | 23 | 15 | 8 |
| *Panurginus cressoniellus* Cockerell, 1898 | 252 | 21 | 231 |
| *Panurginus* sp. 1 | 1 | 1 | 0 |
| *Panurginus* sp. 2 | 1 | 0 | 1 |
| *Protandrena bakeri* Cockerell, 1896 | 23 | 1 | 22 |
| *Protandrena* sp. 1 | 9 | 8 | 1 |
| *Protandrena* sp. 2 | 2 | 2 | 0 |
| *Sphecodes* spp. | 94 | 88 | 6 |
| *Stelis foederalis* Smith, 1854 | 8 | 4 | 4 |
| *Stelis montana* Cresson 1864 | 3 | 0 | 3 |
| *Stelis nitida* Cresson, 1878 | 7 | 0 | 7 |
| *Stelis pavonina* Cockerell, 1908 | 4 | 3 | 1 |
| *Stelis subcaerulea* Cresson, 1878 | 5 | 4 | 1 |

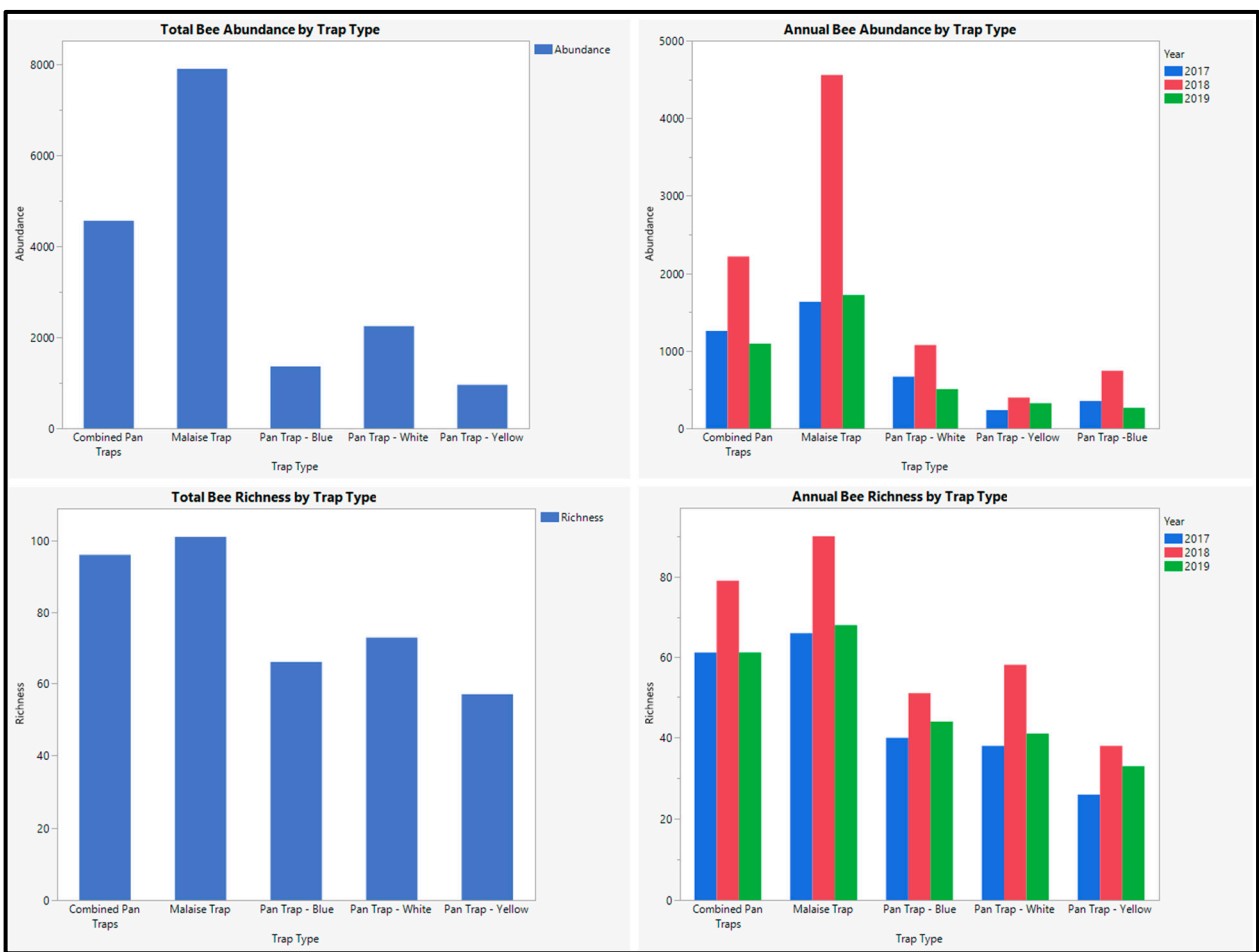

**Figure 3.** The abundance and richness of all bees collected in Malaise and pan traps deployed in the summer months of 2017, 2018, and 2019 in subalpine meadow communities across the Wasatch Plateau in the Manti-La Sal National Forest. Charts are separated by trapping type (Malaise trap and pan trap) and by color (blue, white, and yellow), presented as overall totals (**top left** and **bottom left**) and separated by year (**top right** and **bottom right**).

When assessed as individual colors, the pan traps captured fewer individuals and less richness than both the Malaise traps and the combined pan traps (*p*-values < 0.0001). White pan traps outperformed yellow and blue pan traps in terms of average annual abundance per trapping site (*p*-values < 0.0001 and 0.0242, respectively). In terms of average annual species richness, differences between white and blue pan traps were insignificant; however, white pan traps captured on average 3.8 more species than yellow pan traps per trapping site annually, which was significant (*p*-value: 0.0494).

A species accumulation curve was constructed using observed values (Sobs) and a variety of predictive models (Chao 1, Chao 2, Jacknife 1, Jacknife 2, Bootstrap, MM, UGE) to observe species richness as a function of sampling effort. This was performed for each method individually as well as the total for both Malaise traps and pan traps combined (Figure 4). The rate of species capture exhibited a marked acceleration within the initial 25 samples, after which the rate of novel species accumulation per collection exhibited a discernible decline. Despite this reduction, the accumulation curve for observed richness, as well as for all predictive models, did not plateau, indicating a continuous species discovery. Similar results were observed for Malaise traps and pan traps when assessed individually.

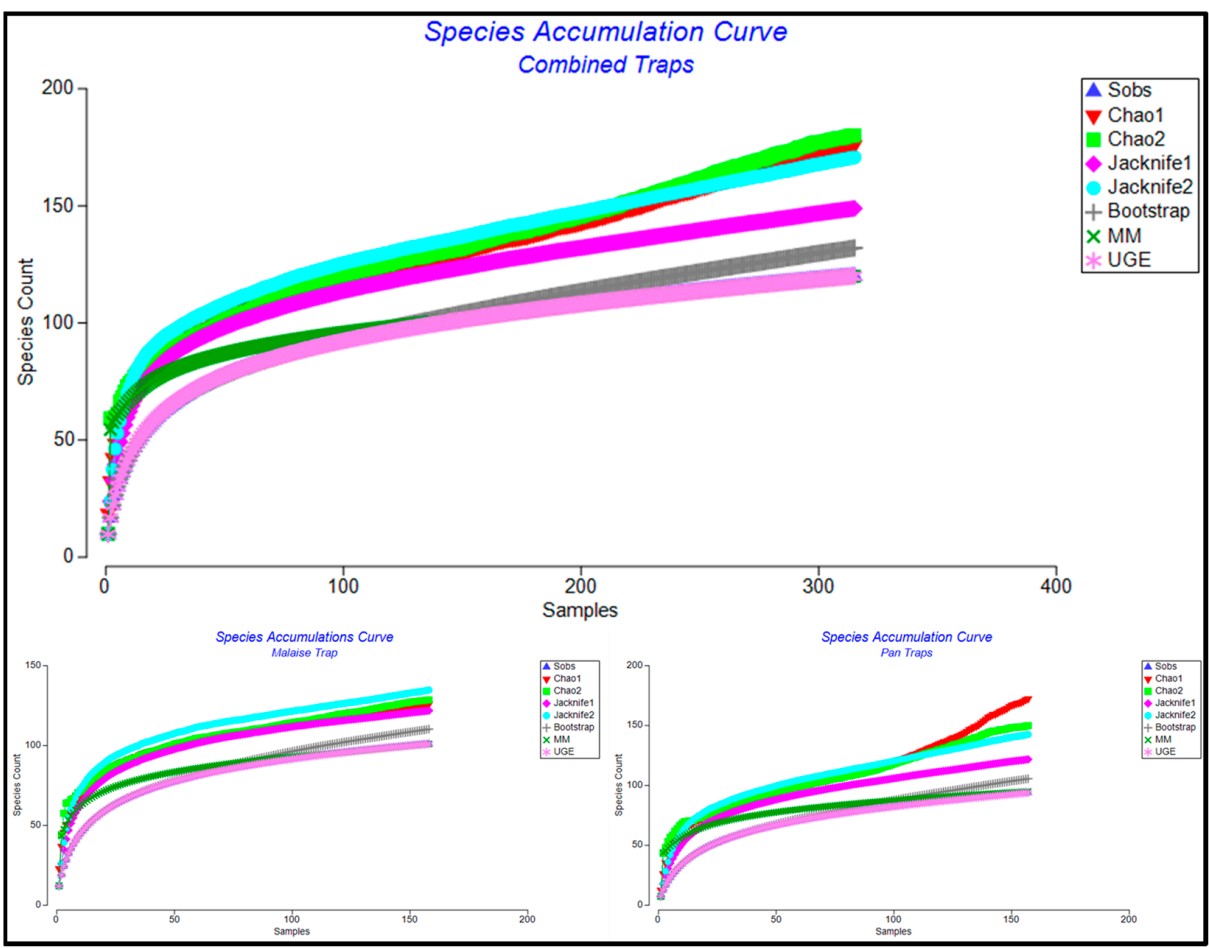

**Figure 4.** Species accumulation curves displaying a variety of modeling techniques to describe community richness as a function of trapping effort. In all graphs, Sobs is directly occluded UGE. The Combined Traps' accumulation curve represents both Malaise and pan trap totals combined. Each trapping method is then modeled individually, Malaise traps (**bottom left**) and pan traps (**bottom right**).

Of the 119 species collected in this study, some were encountered often while others were encountered just a single time. Prevalent genera were determined using a similar method for plant communities by Curtis [29] by calculating the average number of bee genera collected per Malaise trap sample and pan trap sample, which were 5.7 and 5.0, respectively. Each genus was then ranked according to frequency. The six most frequently encountered genera overall were then considered the prevalent genera. These six top groups in order of relative abundance were *Bombus*, *Hylaeus*, *Lasioglossum*, *Osmia*, *Andrena*, and *Megachile*, and accounted for 89.7% of total bee captures. The sampling effectiveness and composition rates for each of these groups were compared for each of the trapping methods.

The composition and abundance of wild bees sampled between the collection methods varied greatly (Figures 5 and 6). Each method had one major contributing genus group, *Bombus* for Malaise traps and *Hylaeus* for pan traps. When broken down by individual pan trap color, there were not large discrepancies in the proportional composition. However, the abundance of specimens captured differs overall by color. Throughout the study, white pan traps collected 2248 specimens, while blue traps captured 1362 specimens and yellow captured 958 specimens (Figure 6).

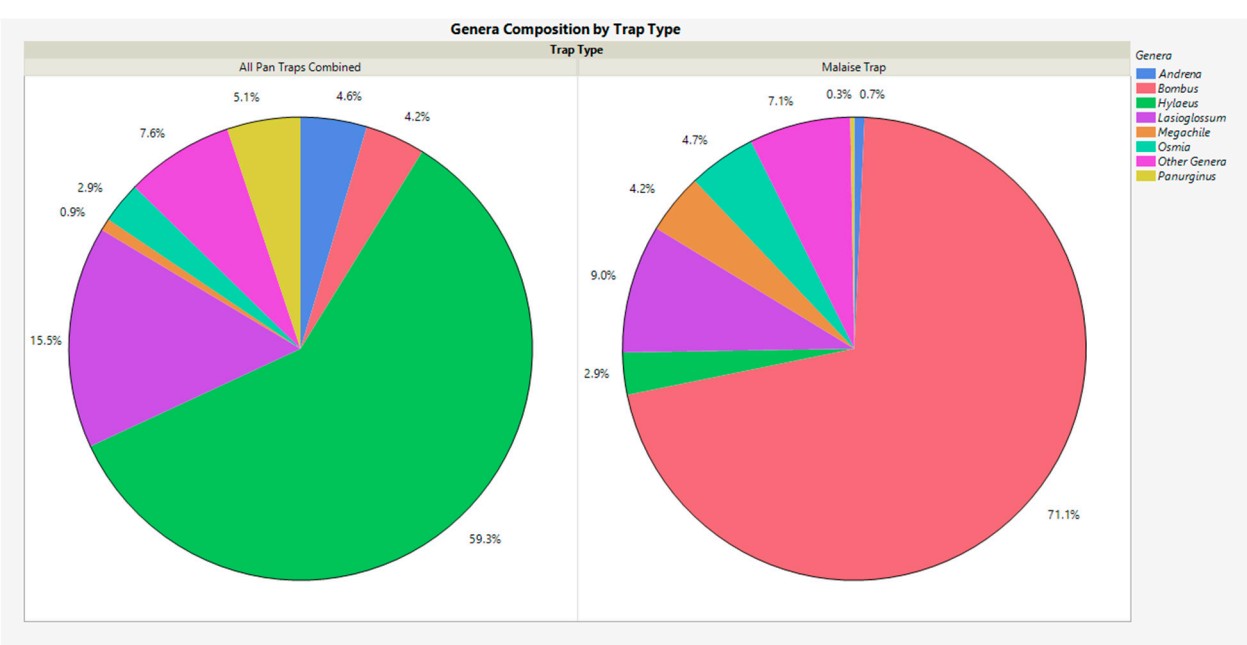

**Figure 5.** The overall composition of genera collected in Malaise traps and pan traps deployed in the summer months of 2017, 2018, and 2019 in subalpine meadow communities across the Wasatch Plateau in the Manti-La Sal National Forest.

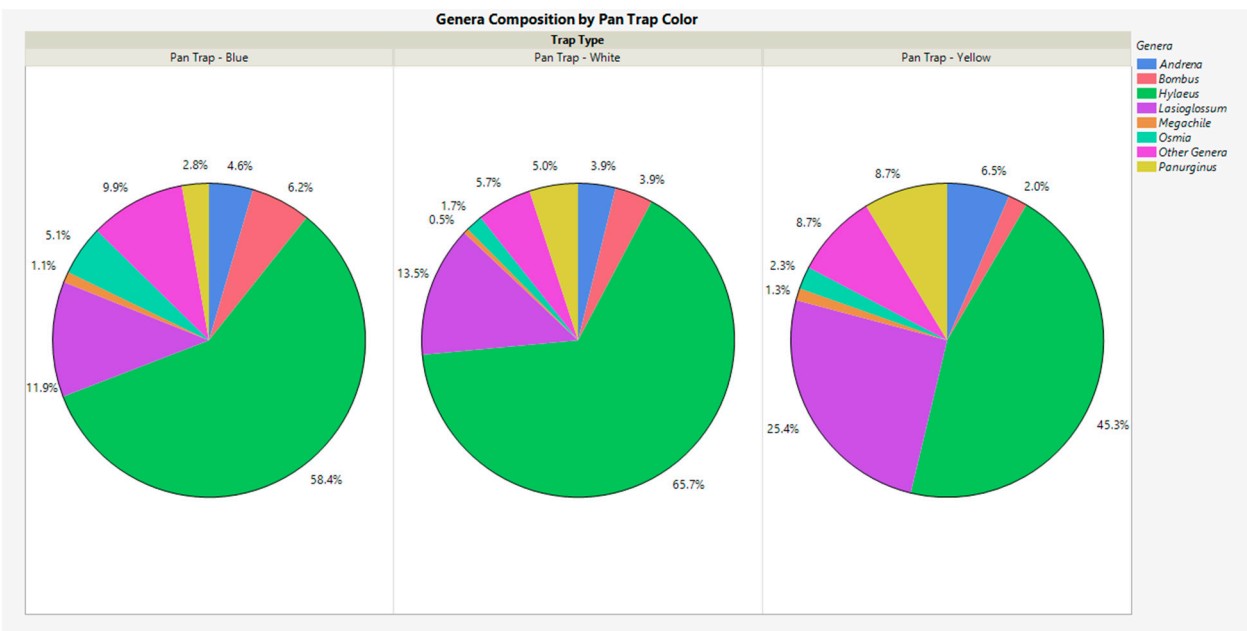

**Figure 6.** The composition of genera collected in blue, white, and yellow pan traps deployed in the summer months of 2017, 2018, and 2019 in subalpine meadow communities across the Wasatch Plateau in the Manti-La Sal National Forest.

The Bombus genus was sampled effectively using Malaise traps capturing 5625 individual specimens. Pan traps were less effective, catching only 191 individuals. All 14 Bombus species observed throughout the course of this study were observed in Malaise trap captures. Four species, *B. fernaldae*, Franklin, 1911; *B. huntii*, Greene, 1860; *B. melanopygus*, Cresson, 1878; and *B. morrisoni*, Cresson, 1874 were captured exclusively by Malaise traps, while the other ten species were captured by both methods. Malaise traps captured on average 8.2 species annually per trap, significantly more than the combined pan traps,

2.2 species (*p*-value < 0.0001). Additionally, each trapping site collected an average of 170.5 individuals annually using Malaise traps while the combined pan trap averaged 5.8 individuals annually (*p*-value < 0.0001; Figure 7).

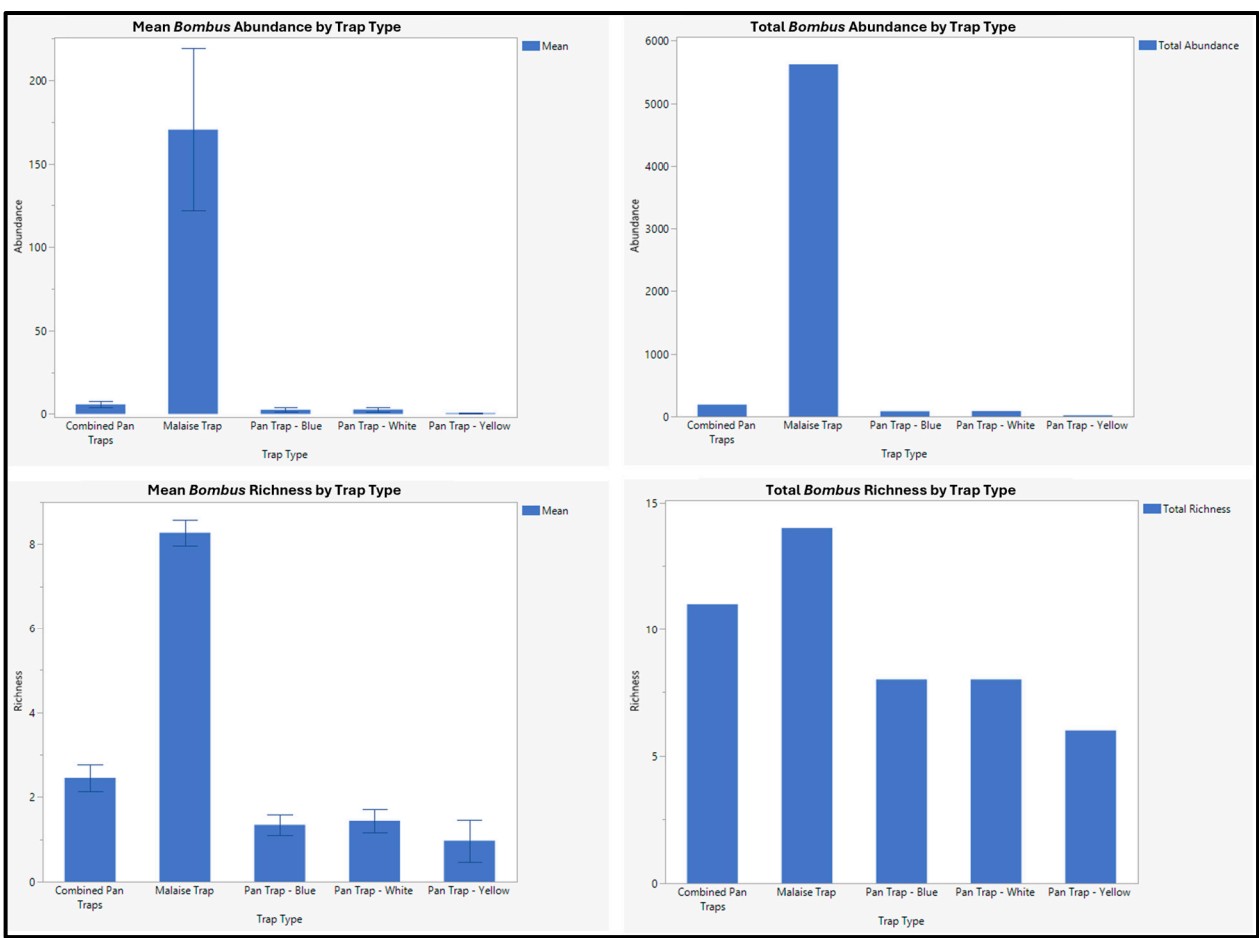

**Figure 7.** Abundance and richness of Bombus spp. collected in Malaise and pan traps deployed in the summer months of 2017, 2018, and 2019 in subalpine meadow communities across the Wasatch Plateau in the Manti-La Sal National Forest. Charts are separated by trapping type (Malaise trap and pan trap) and by color (blue, white, and yellow), and presented as annual mean captures per trap including standard error bars (**top left** and **bottom left**) and as overall summed abundance and richness (**top right** and **bottom right**).

Similar to *Bombus*, *Megachile* richness was completely represented within Malaise trap samples. Pan traps collected eight of the nine different species documented in this study, missing only *M. nevadensis*, Cresson, 1879. While overall richness collected was similar, average annual richness per trapping site was significantly lower with pan traps than Malaise traps (*p*-value < 0.0001) with pan traps capturing less than one *Megachile* species per trapping site annually. In terms of abundance, Malaise traps significantly outperformed pan traps, capturing an average of nearly 10 times the number of *Megachile* per trapping site (*p*-value < 0.0001). Overall, Malaise traps captured 332 individual specimens while pan traps caught just 39 (Figure 8).

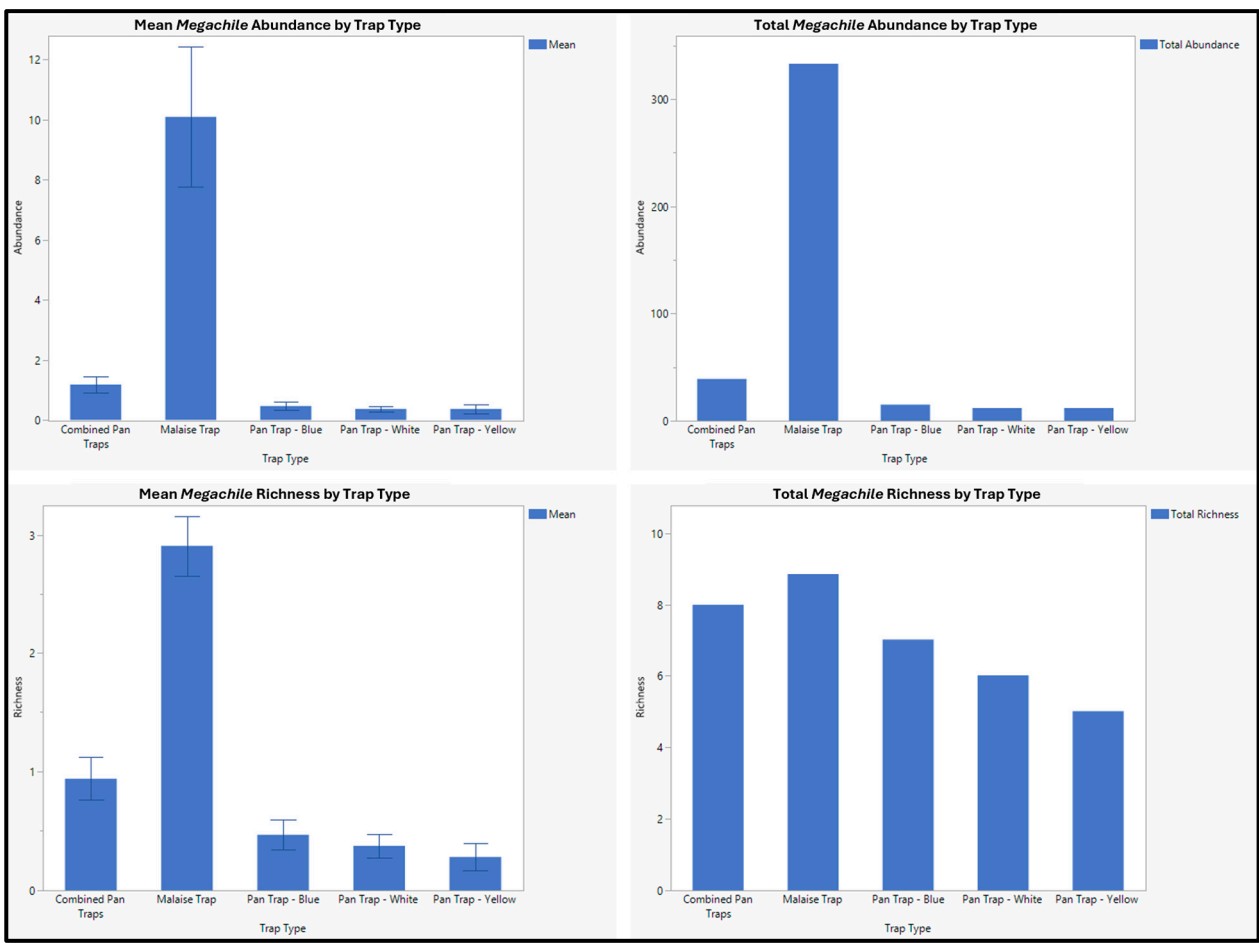

**Figure 8.** Abundance and richness of *Megachile* spp. collected in Malaise and pan traps deployed in the summer months of 2017, 2018, and 2019 in subalpine meadow communities across the Wasatch Plateau in the Manti-La Sal National Forest. Charts are separated by trapping type (Malaise trap and pan trap) and by color (blue, white, and yellow), and presented as annual mean captures per trap including standard error bars (**top left** and **bottom left**) and as overall summed abundance and richness (**top right** and **bottom right**).

*Osmia* followed a similar trend to *Megachile* (Figure 9), although Malaise trapping did not reflect 100 percent of the richness that was identified overall. Each method produced one species that the other method did not. *Osmia raritatis* Michener, 1957 was captured exclusively in pan traps and *Osmia coloradensis* Cresson, 1878 was captured exclusively by Malaise traps. Overall, 16 different *Osmia* species were captured in this study. On average, Malaise traps caught 4.2 species per trapping site annually and pan traps caught 2.3 species, representing a significant difference ($p$-value < 0.0001). Malaise traps had significantly greater success in sampling *Osmia* abundance over the 3 years of this study ($p$-value < 0.0001), collecting on average 11.3 individuals per trapping site annually while pan traps collected 3.9. Overall, Malaise traps captured 374 individual specimens while pan traps captured 131.

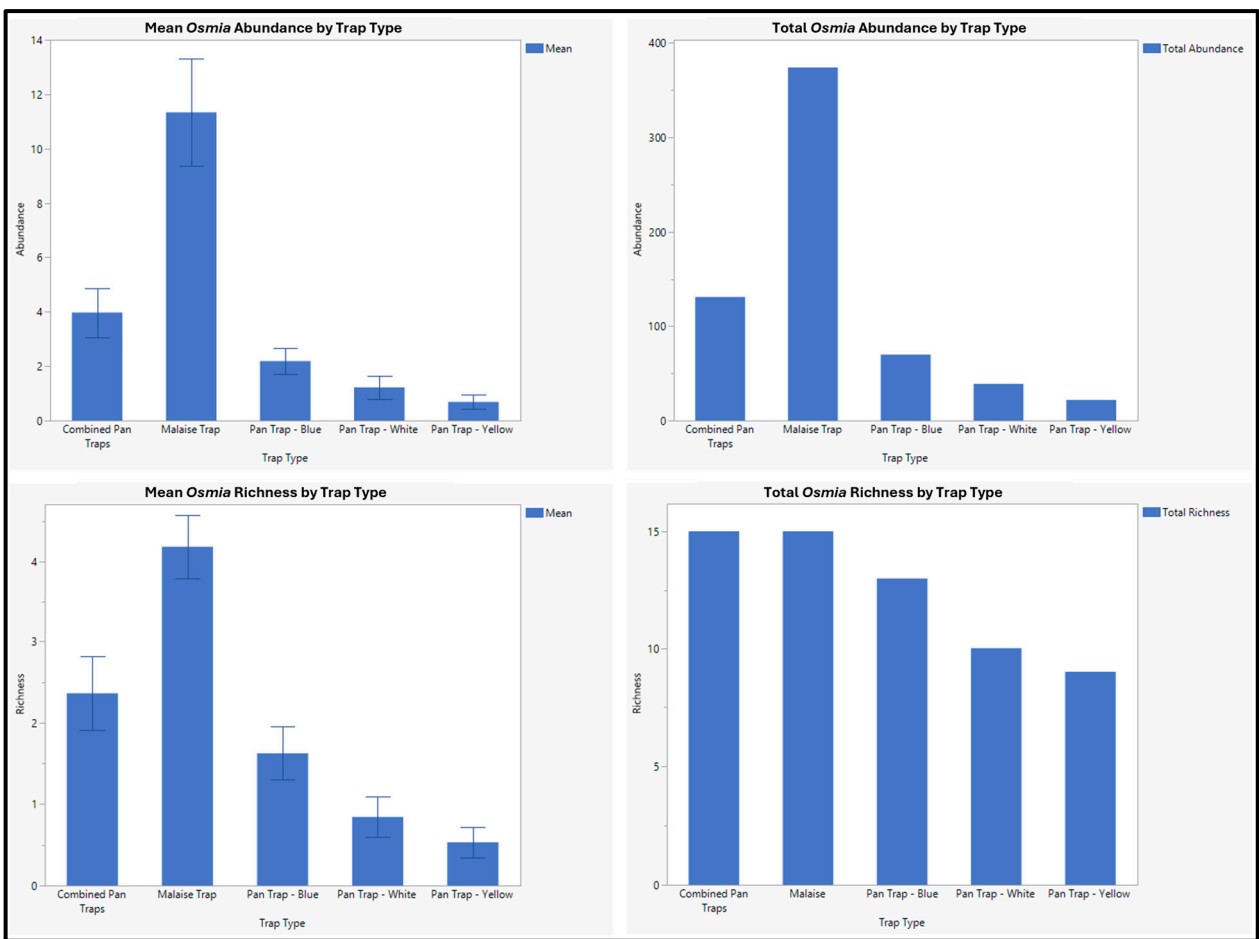

**Figure 9.** Abundance and richness of *Osmia* spp. collected in Malaise and pan traps deployed in the summer months of 2017, 2018, and 2019 in subalpine meadow communities across the Wasatch Plateau in the Manti-La Sal National Forest. Charts are separated by trapping type (Malaise trap and pan trap) and by color (blue, white, and yellow), and presented as annual mean captures per trap including standard error bars (**top left** and **bottom left**) and as overall summed abundance and richness (**top right** and **bottom right**).

In the *Andrena* group (Figure 10), Malaise and pan traps each collected 13 species; however, there were six species captured by Malaise traps that were not captured by pan traps (*A. cerasifolii*, Cockerell, 1896; *A. surda*, Cockerell, 1910; *A. thaspii*, Graenicher, 1903; and three other distinct morphospecies) and six species captured by pan traps that were not captured by Malaise traps (*A. colletina*, Cockerell, 1906; *A. miranda*, Smith 1879; and four other distinct morphospecies that were considerably small in size). While both methods collected the same number of species overall, pan traps on average captured significantly more species per trapping site annually than Malaise traps (*p*-value: 0.0003). While differences between colors were not significant, the blue pan traps sampled only 5 of the 19 *Andrena* species, while the other two individual pan trap colors each captured roughly half of the total richness. The combined pan trap samples significantly outperformed Malaise traps in sampling abundance (*p*-value < 0.0001). Overall, pan traps captured 211 individual specimens and Malaise traps captured 56, equaling roughly 5 additional individuals per trapping site each year. While the differences were insignificant, each of the individual pan trap colors on their own sampled *Andrena* more abundantly than Malaise traps.

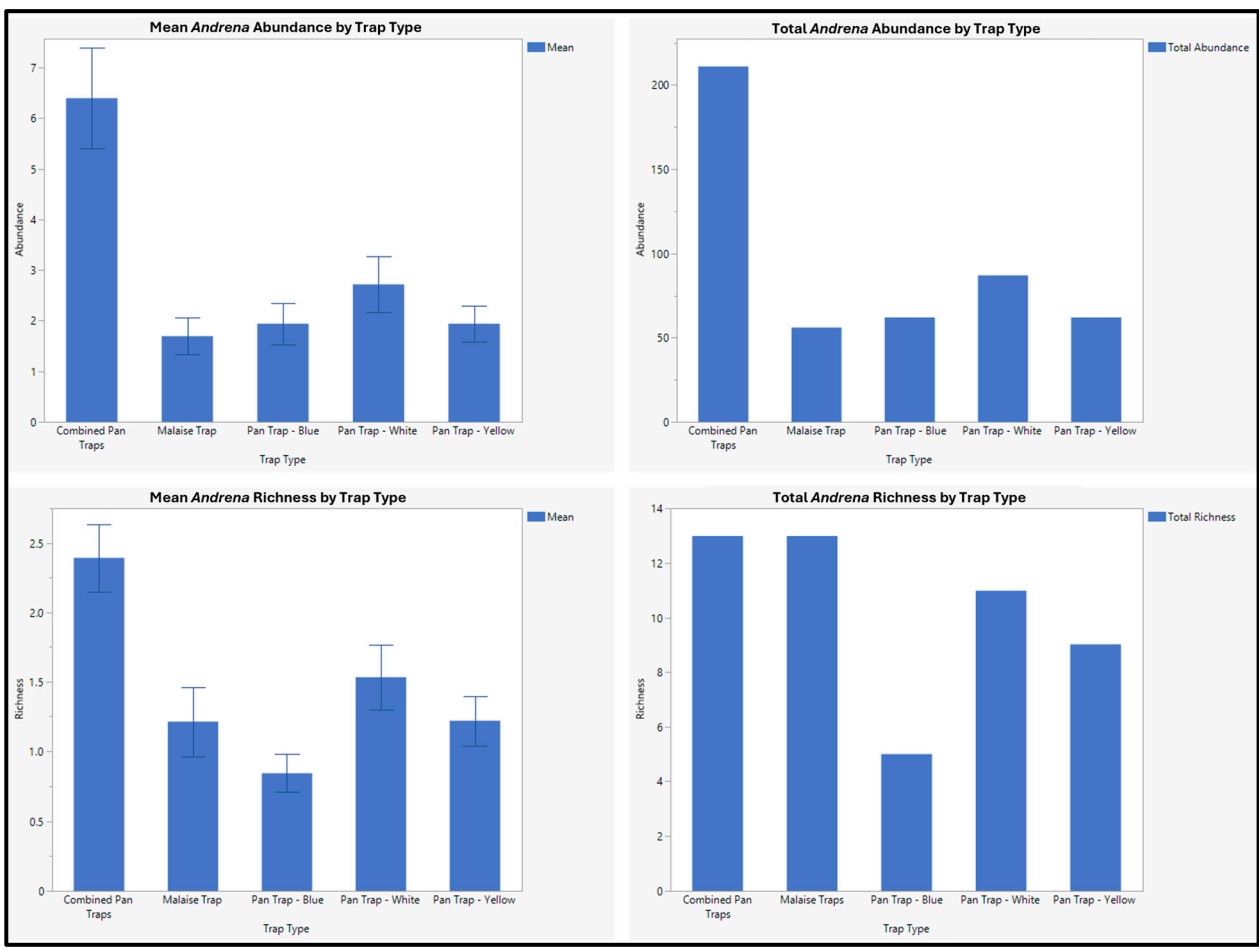

**Figure 10.** Abundance and richness of *Andrena* spp. collected in Malaise and pan traps deployed in the summer months of 2017, 2018, and 2019 in subalpine meadow communities across the Wasatch Plateau in the Manti-La Sal National Forest. Charts are separated by trapping type (Malaise trap and pan trap) and by color (blue, white, and yellow), and presented as annual mean captures per trap including standard error bars (**top left** and **bottom left**) and as overall summed abundance and richness (**top right** and **bottom right**).

Annual trap averages in *Hylaeus* abundance and richness were significantly greater in combined pan trap samples when compared to the Malaise traps (*p*-values < 0.0001). Overall, however, Malaise traps still produced five of the six documented *Hylaeus* species in this study, missing only one morphospecies. On average, pan traps collected approximately three times the number of individuals at each trapping site annually and captured over ten times the number of individuals overall in this study, 2708 individuals to 203 individuals. No singular pan trap color significantly outperformed any of the others in terms of abundance or richness sampling (Figure 11).

Annual capture rates for *Lasioglossum* were not statistically different between Malaise and pan trapping (Figure 12), each capturing on average roughly 22 individuals per trapping site across a year (*p*-value: 0.9952). Likewise, each method caught on average two to three species/subgenera annually per trapping site; these differences in richness were not statistically significant (*p*-value: 0.0776). *Lasioglossum s. str.* sp. and one *Lasioglossum* morphospecies were found exclusively in pan traps. Malaise traps did not collect any unique *Lasioglossum* species/subgenera. Overall, each method produced a similar number of individual specimens, Malaise traps at 709 and pan traps at 708. Due to the limited identification of *Lasioglossum* in this study, the richness is notably underreported for both trapping methods. Differences in richness between each method may vary if all individuals were to be identified beyond the subgenus level.

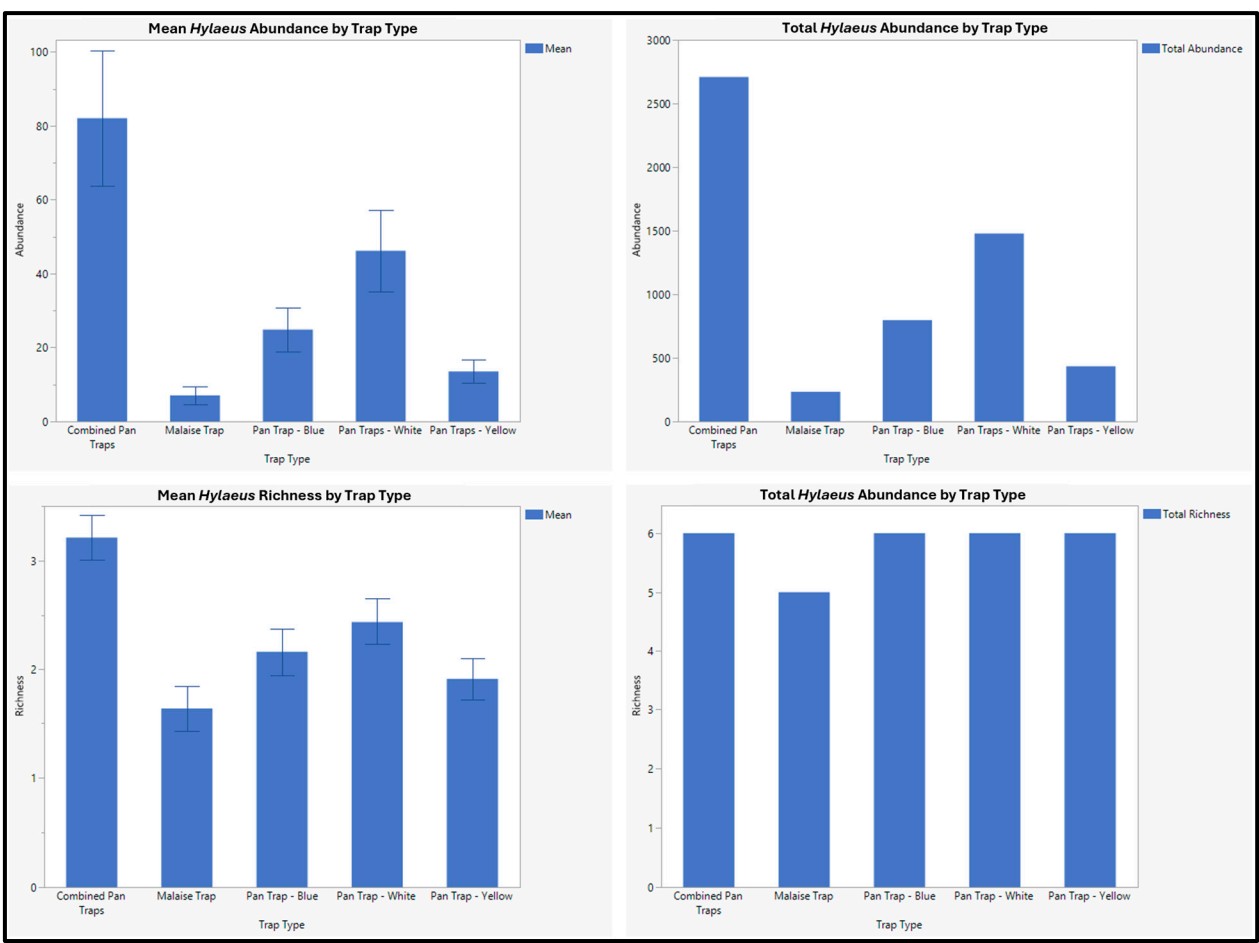

**Figure 11.** Abundance and richness of *Hylaeus* spp. collected in Malaise and pan traps deployed in the summer months of 2017, 2018, and 2019 in subalpine meadow communities across the Wasatch Plateau in the Manti-La Sal National Forest. Charts are separated by trapping type (Malaise trap and pan trap) and by color (blue, white, and yellow), and presented as annual mean captures per trap including standard error bars (**top left** and **bottom left**) and as overall summed abundance and richness (**top right** and **bottom right**).

To better understand broader bee group dynamics, bees were organized into subfamilies and the trapping methods for these groups were assessed for similarity using a PCA analysis (Figure 13). Dimensions 1 and 2 express 34.6% of the variability in the dataset. While direct metrics of bee size were not recorded or compared, the two dimensions that describe the greatest amount of variability appear to be related to bee size. Pan trap captures centered around bee subfamilies with many individuals of small body size (e.g., Hylaeinae and Panurginae, and Halictinae). The genera in our samples associated with these groups, primarily *Hylaeus*, *Panurginus*, and *Lasioglossum*, range from 3 to 10 mm in most cases and constitute more than 80% of the pan trap sample. Pan traps appear relatively less successful at capturing large-bodied bees such as those belonging to subfamilies of Megachilinae, and Apinae, especially *Bombus* (Figure 14). Malaise traps appear relatively more successful at catching groups associated with larger-bodied bees such as *Bombus*, *Megachile*, and *Osmia*, which most cases range between 7 and 20 mm and constitute 80% of the Malaise trap samples. The heavy overlap of data between individual pan trap colors suggests that there is a strong similarity between each individual pan trap color and the bee groups they collect (Figure 13).

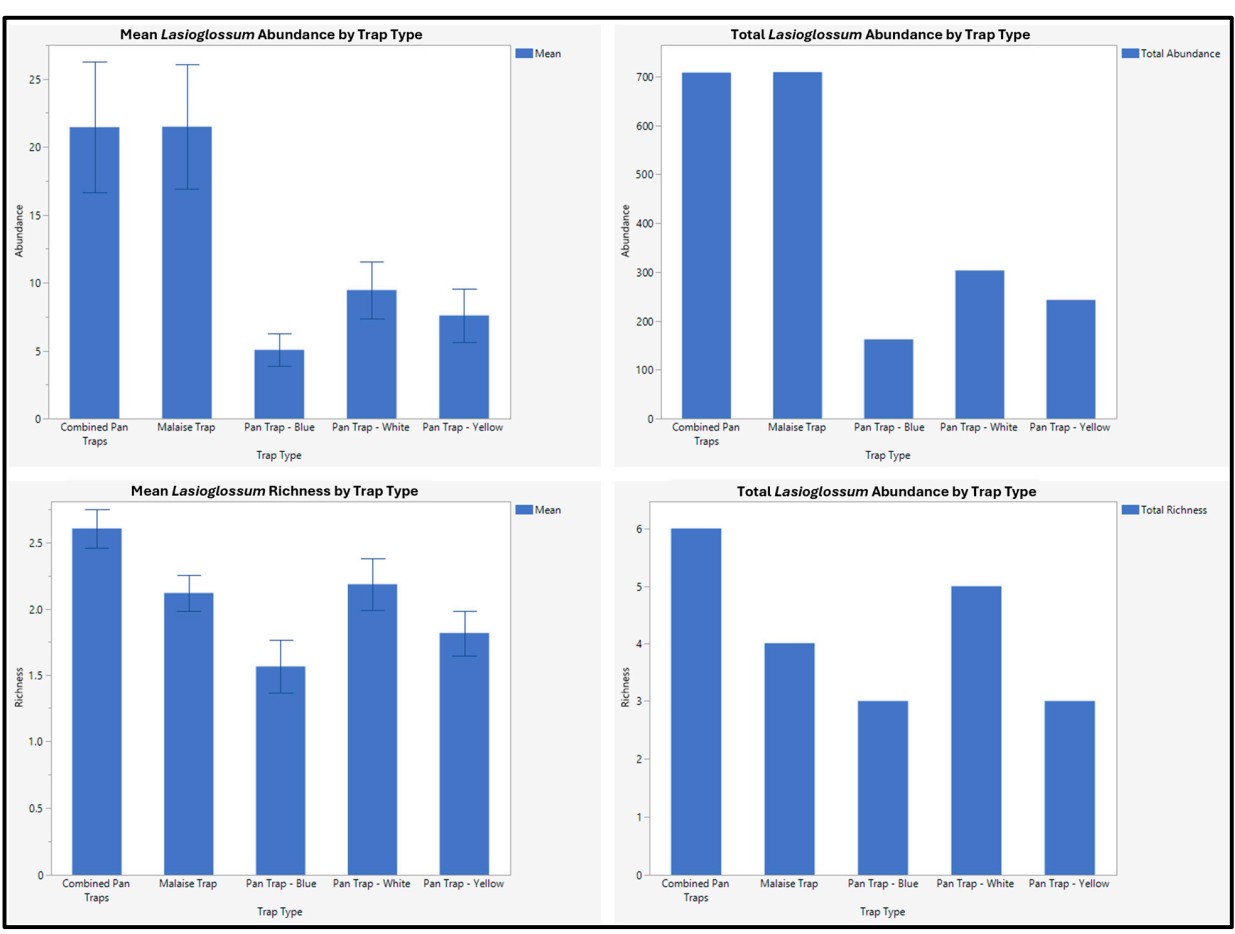

**Figure 12.** Abundance and richness of *Lasioglossum* spp. collected in Malaise and pan traps deployed in the summer months of 2017, 2018, and 2019 in subalpine meadow communities across the Wasatch Plateau in the Manti-La Sal National Forest. Charts are separated by trapping type (Malaise trap and pan trap) and by color (blue, white, and yellow), and presented as annual mean captures per trap including standard error bars (**top left** and **bottom left**) and as overall summed abundance and richness (**top right** and **bottom right**).

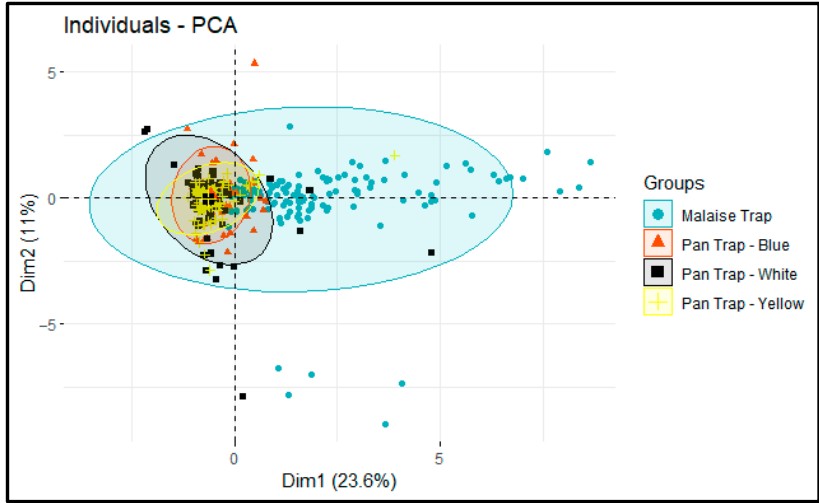

**Figure 13.** The PCA analysis of bees captured in the summer months of 2017, 2018, and 2019 in subalpine meadow communities across the Wasatch Plateau in the Manti-La Sal National Forest, which assessed the similarity of bee subfamily abundance captures between Malaise traps and each of the pan trap colors.

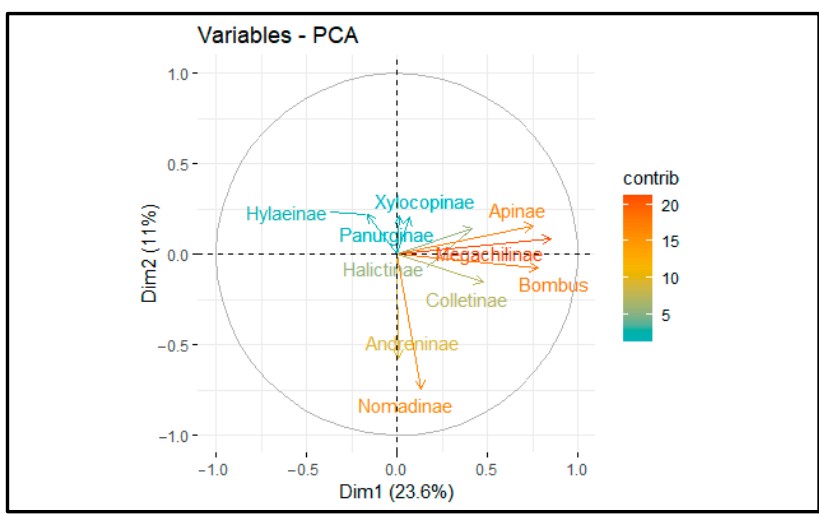

**Figure 14.** A PCA analysis variable overlay of Figure 13, displaying bee subfamilies (Bombus was added independent of the rest of Apinae due to its outsized influence on the group as a whole).

## 4. Discussion

Studies evaluating the trapping effectiveness for these two methods are common in the published literature and often yield conflicting results in different environments. Geroff et al. [30] found that Malaise traps capture the greatest richness and abundance of bees compared to pan trapping in open prairies, with blue pans being the most successful of the pan trap colors. In stark contrast, Krahner et al. [22] found pan traps to be the most effective method of species capture in both richness and abundance in agroecosystems. Additionally, they found that Malaise trap captures were so dismal that they recommend against using these traps as a standard monitoring methodology all together. In closed forest systems, McCravy et al. [19] likewise reported that pan traps produce the greatest abundance and richness of bees with yellow being the most successful color and Malaise traps capturing only 17 individuals. Bartholomew and Prowell [31] did not find a significant difference between richness captured by either of the two methods but experienced a significantly greater abundance of bees captured within pan traps. Campbell and Hanula [18] sampled for bees in three different forested systems and found that pan traps had the greatest richness and abundance of bees, outperforming both Malaise traps and Malaise traps with colored panels. They reported that blue and white pan traps were the most successful colors used. Other studies that did not look directly at the comparison between Malaise trapping but used pan trapping as a method have mixed results when comparing pan trap color. Archarya et al. [32] found that blue pan traps had the highest rates of bee capture and species accumulation in livestock pastures, while Munyuli [33] found that yellow pan traps were the most effective at capturing bee abundance, but that blue pan traps captured the greatest diversity of bee species compared to yellow and white.

Our results contribute to the mixed conclusions provided by these studies and other studies. Over the course of the three years, a greater abundance of bees was collected using Malaise traps. Individual pan trap colors alone captured a significantly fewer number of species and individuals than Malaise traps; however, their aggregate richness was not statistically significant from Malaise trap samples across years. Within the context of specific colors, white pan traps resulted in the greatest number of species captured; however, no single color was statistically superior to any other in terms of richness. Each color was valuable as each contributed unique species to the pan trap collections made throughout this study. Likewise, both Malaise and pan traps contributed unique species to the overall collection effort. Thus, the bee community was better represented by the utilization of both trapping methods. Studies often found considerable rates of overlap between Malaise and pan trapping [18] and that the addition of hand netting improved richness to pan trapping alone [33–35]. With each additional method, there appears to be a greater likelihood of

improving richness. Joshi et al. [36] evaluated capture efficiency in pan traps, vane traps, Malaise traps, and hand netting and found that 48% of bee species were captured by a single collecting method. While this was not reflected in such great measure in this study, there was increased richness with additional trap types and colors. Because the species accumulation curve never completely plateaued, we may infer that additional time or methods such as hand netting may have improved our community sampling; however, there is a diminished return by repeated sampling. While additional methods appear to improve richness, they require additional effort, which must be balanced across the time and budget available for a project.

Beyond the observations of this study, others suggest that Malaise traps are particularly effective at capturing a greater abundance of bees than pan traps when deployed in open systems such as meadows and prairies [30]. In these systems, light is abundant as opposed to a forested system with a closed canopy where pan traps have been documented collecting greater abundances and Malaise trap capture was dismal [18,31]. McCravy et al. [19] propose that Malaise trap success relies on insect phototaxis and explains that the outsized success of Malaise trap captures in open systems may be due to a greater abundance of light. Our sampling data offer further support of this hypothesis as it largely reflects the findings of Geroff et al., [30] in contrast to other studies in closed forested environments.

In addition to extra light, the abundance of floral cover is very high in subalpine meadows. Westerberg et al. [37] suggest that pan traps are negatively biased to social bees, e.g., bumble bees and honey bees, when there is a higher abundance of flowers nearby. Social bees have the capacity to communicate with one another and as flower constants, they often forage on one species of plant so long as there is a profitable nectar reward [38]. This phenomenon could result in a scenario where a specific floral abundant meadow attracts a greater number of social bees that are more likely to be randomly intercepted by a Malaise trap in that meadow while simultaneously less likely to attempt foraging on pan traps. This may explain part of the outsized capture rate of bumble bees in Malaise traps that we observed in our samples. Westerberg et al. [37] found no such negative bias for solitary bees, which would also be consistent with our observations.

Joshi et al. [36] observed that larger-bodied bees such as bumble bees were negatively biased in pan trap captures as they were big enough and strong enough in some instances to escape. This observation held true in several observed instances during our field collections as bumble bees were seen swimming to the bowls' edge and sometimes being able to climb out even after capture. Larger-bodied bees were more frequently associated with Malaise traps as opposed to pan traps. A negative bias toward larger-bodied bees was seen in our PCA analysis with a variable overlay that skewed Malaise trap captures toward Apinae and Megachilinae, which are typically larger than the other subgroups that were compared.

Size, however, is not the only variable associated with trapping success. *Lasioglossum* species are small and seem to be captured in similar numbers between the two methods. However, equally as small, *Hylaeus* species show a much higher capture rate in pan traps as opposed to Malaise traps and are captured at rates roughly 11 times higher in pan traps.

When attempting to understand population dynamics, abundance is a crucial metric. In order to observe and make comparisons through statistical models, it is imperative to have an appropriate volume of specimens to assess. Depending on the species of interest, a determination must be made as to which trapping method is most appropriate as the volume, i.e., abundance, of individuals caught is not consistent between methods.

For a variety of reasons, one of these methods or the other may be better suited to a specific project. On these subalpine meadows of the central Wasatch Plateau, we encountered several limitations and benefits for each method. Malaise traps have the advantage of remaining active for the entire floral season, only requiring weekly collection and maintenance. After the trap is deployed, bottle replacement takes just a few minutes. They are low-maintenance traps, require little field time, and can produce robust datasets. Several drawbacks include an abundance of bycatch, initial trap cost relative to other trapping methods, and visibility to the public eye, which invites curiosity and potential

disturbance. They are particularly effective in capturing bumble bees and other large-bodied bees. Pan traps are relatively discrete, inexpensive, and easy to deploy, and produce far less bycatch than Malaise traps. However, during the growing season, they cannot be reasonably left active for long periods of time due to sporadic weather events and sensitivity to wildlife and livestock disturbance. There are several species that have a seemingly high allure to these traps such as *Hylaeus*; however, larger insects can more easily escape.

Rare species, or those with smaller population sizes, may be less resilient to perturbations in an environment [39] and as such are invaluable when assessing community trends. Both methods captured the same number of these less frequently sampled species. Malaise traps captured 19 individual species that were present only once or twice in the dataset and pan traps captured 19, indicating that within this community, the sampling contribution from each method was invaluable in assessing richness and diversity.

## 5. Conclusions

After three years of collections, Malaise and pan traps yielded very similar overall species richness despite Malaise traps sampling significantly more bee richness per trap annually. The richness overlap between the two sampling methods was 65 percent and each method collected a similar number of rare species. Vastly different abundances of bees were collected by one method or the other depending on the species, but in general, Malaise traps captured a greater abundance of larger bees and pan traps smaller bees. Depending on the aims of a particular study, either method could yield beneficial results and serve as a primary or supplemental passive collection strategy. In this case, we found that both trapping methods were important in understanding bee community composition.

Pollinators are recognized as critical and integral elements of healthy environments. There could be many reasons for the collection of pollinators, perhaps to study a specific taxon, understand specific plant pollinator interactions, create baseline community datasets, or simply collect as a hobby. Understanding collection method efficiencies in different environments can help practitioners and researchers better achieve their goals. Differing results for these methodologies exist across different geographical locations, habitats, and even individual species within their communities. Each of these variables presents unique challenges and opportunities that dictate trapping success. Therefore, a one-size-fits-all approach may not be appropriate in every situation. The abundance of the literature on the subject should help researchers better assess which methods are most appropriate for their work.

**Author Contributions:** Conceptualization, N.A., V.A. and S.P.; methodology, N.A. and V.A.; software, N.A. and S.P.; validation, S.P., N.A. and V.A.; formal analysis, N.A. and T.T.; investigation, N.A., J.K. and D.L.; resources, S.P., R.J. and V.A.; data curation, N.A., J.K. and D.L.; writing—original draft preparation, N.A.; writing—review and editing, N.A., J.K., S.P., V.A. and R.J.; visualization, N.A. and T.T.; supervision, S.P. and V.A.; project administration, N.A., S.P. and V.A.; funding acquisition, V.A. and R.J. All authors have read and agreed to the published version of the manuscript.

**Funding:** This research was funded by The United States Forest Service, grant number 16-CS-11041000-022. Funding was supplemented internally by Brigham Young University.

**Data Availability Statement:** The data presented in this study are publicly available on Figshare and can be found by doi: 10.6084/m9.figshare.26045320.

**Acknowledgments:** We would like to acknowledge the Monte L. Bean Museum and their associates, specifically Shawn Clark, as well as the USDA Pollinating Insect-Biology, Management, Systematics Research unit in Logan, Utah, specifically Terry Griswold and Harold Ikerd for their help in identifying bee specimens.

**Conflicts of Interest:** The authors declare no conflicts of interest. The funders had no role in the design of the study; in the collection, analyses, or interpretation of data; in the writing of the manuscript; or in the decision to publish the results.

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
