# Peer review of "Pan Trapping and Malaise Trapping: A Comparison of Bee Collection Techniques in Subalpine Meadows"

_diversity, doi:10.3390/d16090536_

Round 1

Reviewer 1 Report

Comments and Suggestions for Authors

This study examines the relative value of pan traps of different colors and Malaise traps in capturing the bee richness and diversity of a community. The study ran from July to September and spanned 3 years and 16 sites. The study provides a thorough test of the different methods and the conclusions are well supported. As noted in the Discussion, the results from this study differed from those of previous studies in terms of whether pan traps or Malaise traps had the greater bee abundance and which pan trap color captured more bees. As such, the study provides useful information showing that the community type, e.g. agroecosystem, prairies etc., makes a difference as to the effectiveness of different trap types. It also emphasizes the need to employ multiple methods to capture the breadth of species diversity. It does leave open the question of how to combine the results from different trap methods to characterize the community as a whole. But it also points out which trap method is best if your goal is not to examine the entire bee assemblage but focus on a species group, e.g. Bombus.

Edits

There should be capital m for Malaise

19 and an array of blue, white and yellow pan traps, nine at each site.

Figures 6-11 The name of the group, Bombus etc., should be in the figure itself, not just the legend. This is done partially in Fig. 6 for Bombus.

Author Response

Dear Reviewer,

Thank you for your time and thorough review of our work. The comments and feedback you have given us has helped us refine and enhance this article for which we are very appreciative. The specific edits you provided helped us understand what items specifically we needed to clarify or amend. Each addressed below:

There should be capital m for Malaise – Found and remedied throughout

19 and an array of blue, white and yellow pan traps, nine at each site. – Good catch this is more clear.

Figures 6-11 The name of the group, Bombus etc., should be in the figure itself, not just the legend. This is done partially in Fig. 6 for Bombus. – The figures should be much clearer now, names have been added.

Reviewer 2 Report

Comments and Suggestions for Authors

This is an interesting study comparing sampling bee community composition with pan traps and malaise traps and getting very different result.  It is marred by numerous spelling errors for the names of the subset of bees mentioned, essentially no description of the study habitat or its weather, and an inadequate description of the pan traps.  No overall species list broken down by sex, number of individuals and trap type is presented.  This should be required in any study of this type and I would not recommend publishing this paper without one.

A colored pie graphic dramatically indicates the great diffences in community structure depending on the sampling method, but because of the many small pie slices and their similar colors, its general utility is greatly lessened.

The authors spend considerable space comparing catch results for the pan traps of different colors but as they give no details about the actual traps, the value of this is greatly lessened.

Author Response

Thank you for your careful and comprehensive review of our paper.  Following your suggestions, we have made significant revisions to the paper addressing issues for which you had legitimate concerns. Specific revisions include

  1. Spelling errors have been corrected for bee names throughout the paper.
  2. A more comprehensive description of the study sites have been provided, including information of plant community, elevations, climate, etc.
  3. Pan traps and the trapping array have been described in greater detail.
  4. A complete species list and numbers caught by each trapping method has been included. The sex of bee specimens was not completed as part of the ID process and is thus not available.
  5. The pie charts have been remade to reflect contributions of the perevalent species with contributions of others too small to be clearly visible in the pie charts combined in an other category.
  6. Many of the word suggestions and minor edits suggested throughout the manuscript were also incorporated. Again, thank you for you detailed review.

Reviewer 3 Report

Comments and Suggestions for Authors

The paper is pretty good and will be interesting to readers.  There are some problems with the application of names and acknowledgment of contributors.  If I'm properly understanding the methods as written, then identification of ~75% of the specimens and ~90% of the names was provided by people who are not even acknowledged by name, just by institution.  Unless some agreement was made with those USDA identifiers, they deserve more recognition and maybe even co-authorship.

In most journals, scientific names need to include their author, and sometimes year of publication, with their first mention in the text.  Unless this journal has different rules, this should be fixed throughout the manuscript.  Scientific names should always be italicized, even in figures.  A genus abbreviation should never start a sentence, the genus needs to be spelled out. Some of the scientific names mentioned in the text have typos that should be checked.   

The paper would benefit from a few photographs of the traps themselves in their habitat.

Might be useful to provide a little more rigor or justification for the size bias in trapping style.  At the very least provide a size range for the categories that you compared, or a minimum and maximum body size collected from each trap type, or a citation to a paper where different bee groups are placed into size categories.

Might be useful to cite this paper which shows size bias in malaise vs pan trapping in Mutillidae: 

1.     Vieira, C.R., Waichert, C., Williams, K.A. and Pitts, J.P. , 2017. Evaluation of Malaise and Yellow Pan Trap Performance to Assess Velvet Ant (Hymenoptera: Mutillidae) Diversity in a Neotropical Savanna. Environmental Entomology, 46(2): 353-361.

From my experience, it isn't that malaise traps skew toward a higher size of insect collected, but that they passively collect insects of all sizes.  The pan traps, however, miss out on collection of larger bodied insects, perhaps due to their behavior.  Could be different for bees, though.

Author Response

Dear Reviewer,

Thank you for your time and thorough review of our work. The comments and feedback you have given us has helped us refine and enhance this article for which we are very appreciative. The specific edits you provided helped us understand what items specifically we needed to clarify or amend. These are each addressed below:

The paper is pretty good and will be interesting to readers.  There are some problems with the application of names and acknowledgment of contributors.  If I'm properly understanding the methods as written, then identification of ~75% of the specimens and ~90% of the names was provided by people who are not even acknowledged by name, just by institution.  Unless some agreement was made with those USDA identifiers, they deserve more recognition and maybe even co-authorship. – Thank you for looking out for these gentlemen. We asked them what they would like in the form of acknowledgement, authorship, or compensation before this study began. Their request was only an institutional acknowledgment and a contribution in the form of donated materials and retention of the study specimens as compensation; however, we have included the names of primary contributors to the identification effort in the acknowledgements.

In most journals, scientific names need to include their author, and sometimes year of publication, with their first mention in the text.  Unless this journal has different rules, this should be fixed throughout the manuscript.  Scientific names should always be italicized, even in figures.  A genus abbreviation should never start a sentence, the genus needs to be spelled out. Some of the scientific names mentioned in the text have typos that should be checked.   – These Items were amended throughout

The paper would benefit from a few photographs of the traps themselves in their habitat. – Added 144

Might be useful to provide a little more rigor or justification for the size bias in trapping style.  At the very least provide a size range for the categories that you compared, or a minimum and maximum body size collected from each trap type, or a citation to a paper where different bee groups are placed into size categories. – added additionally clarity and justification 282-292

Thank you again for your thoughtful review and constructive comments.

Round 2

Reviewer 2 Report

Comments and Suggestions for Authors

By expanding the description of the methods and adding Table 2 , the authors have largely dealt with my prior concerns with the manuscript.  Table 2 still has quite a few minor problems but all should be easy to deal with.  The treatment of Lasioglossum remains problematic and almost certainly results in an underestimate of number of species.  This probably cannot be rectified but should be acknowledged.

Author Response

Thank you for the additional comments and review. The following comments have been incorporated throughout:

Table 1.  It not obvious why authors names are used for bees but not plants. - Fixed

Table 2:  Delete all the commas after the specific epithets - Fixed

The author or sp. sp. or spp. should not be italicized - Fixed

Andrena hirticincta, not hiricincta Andrena spp, Atoposmia spp. , Ceratina spp., Coelioxys spp., Diadasia spp., Hylaeus spp, Lasioglossum spp., Melissodes spp., Nomada spp., Panurginus spp., Protandrena spp., Sphecodes spp. all should be sp., this includes spp. 1, spp. 2 which should be sp. 1, sp. 2.... For all these, there is either only one specimen per genus or the species in question is identified by a unique number - Fixed

Dialictus, Evylaeus and Lasioglossum s. str. are subgenera of Lasioglossum and should be in parentheses.  – Fixed

No way to tell but I assume the Dialictus and Evylaeus are actually a mix of species (hence spp.).  Also, I would assume that Evylaeus here is a mix of Evylaeus, Hemihalictus and Sphecodogastra (all subgenera whose members were often included in Evylaeus).  These probably should be Lasioglossum (Dialictus) spp. and Lasioglossum (Evylaeus) spp. unless they refer to single species. If at all possible, it would be useful to indicate what subgenus (or subgenera) Lasioglossum sp. 1 and sp. 2 belong to. - Fixed to the extent possible and a paragraph is added explaining the limited richness overall due to the grouping of subgenra in lasioglossum.

Panurginus cressoniellus, not cressoniellas Stelis pavonina, not Stelis pavovina Please recheck all the names. -Fixed

 You need to recheck the math in the table since a few of the numbers don't agree (Hylaeus annulatus 172 + 2433 does not equal 2617)  -Fixed

line 110: Malaise should not be capitalized unless it starts a sentence.  This is very inconsistently handled throughout the paper.  Why Malaise trap but not Pan trap? – Another reviewer requested Malaise traps be capitalized for their inventor Rene Malaise – this is now capitalized throughout.

190, and Megachile, not and Megachile – Fixed

235 & 237  Here and elsewhere, one does not put a comma between a specific epithet and the species author. – Fixed

235: Michener is misspelled. - Fixed

Since species authors are given in table 2, it is not necessary to repeat them in the text - although that can be an editorial preference. – They are added throughout the text unless requested differently by the journal.

  1. Names of higher categories (families, subfamilies, tribes and so forth) are not italicized so it is Hylaeinae, not Hylaeinae - Fixed

363:  I prefer bees over Anthophila but that's just a personal preference,. Anthophila sounds more technical but it has no formal standing.  Similarly. I prefer honey bee and bumble bee over honeybee and bumblebee since we are talking about bees, not something like butterflies which are not flies. - fixed

Apinae and Megachilinae should not be italicized - fixed